# The Evaluation of a High-Fidelity Simulation Model and Video Instruction Used to Teach Canine Dental Skills to Pre-Clinical Veterinary Students

**DOI:** 10.3390/vetsci10080526

**Published:** 2023-08-16

**Authors:** James Fairs, Anne Conan, Kathleen Yvorchuk-St. Jean, Wade Gingerich, Nicole Abramo, Diane Stahl, Carly Walters, Elpida Artemiou

**Affiliations:** 1Department of Clinical Sciences, Ross University School of Veterinary Medicine, Basseterre P.O. Box 334, Saint Kitts and Nevis; kstejan@gmail.com (K.Y.-S.J.); nabramo@rossvet.edu.kn (N.A.); dstahl@rossvet.edu.kn (D.S.); cwalters@rossvet.edu.kn (C.W.); 2Centre for Applied One Health Research and Policy Advice, City University of Hong Kong, 31 To Yuen Street, Kowloon, Hong Kong SAR 999077, China; ayconan@cityu.edu.hk; 3Pet Dental Center, 9250 Corkscrew, STE 18, Estero, FL 33928, USA; wadeging@yahoo.com; 4Texas Tech University School of Veterinary Medicine, 7671 Evans Dr., Amarillo, TX 79106, USA; elpida.artemiou@ttu.edu

**Keywords:** veterinary dentistry skills, veterinary dental education, veterinary clinical skills, simulation, high-fidelity dental model

## Abstract

**Simple Summary:**

Dental disease is the most diagnosed disease in small-animal general practice and has a significant impact on the health and welfare of patients. Despite the prevalence of dental disease, there is a recognized gap between the dental skill training veterinary students receive and the expectations of employers regarding the competencies of new graduates in this field. Furthermore, there is a lack of published research reporting on veterinary dental skill training. This study evaluates the models and videos used to teach canine dental core skills. Dental skill acquisition and confidence were found to be higher in students who were trained using models rather than videos. However, there was no significant difference in perceptions related to small-animal dentistry between students trained using the different modalities. The authors recommend using both models and videos to train veterinary students in order to optimize skill acquisition in this field. The conclusions drawn from this research can be used to improve student training so that new graduates may enter the profession better prepared to demonstrate these skills.

**Abstract:**

In recent years, there has been an increased focus on the teaching of small-animal dentistry to veterinary students in order to address the recognized gap between dental skill training and the expectations of employers regarding the competencies of new graduates in this field. In this study, third-year veterinary students were trained in three canine dental core skills using either a high-fidelity model (Group A) or video instruction (Group B). An objective structured clinical examination was used to assess skill acquisition and questionnaires were distributed in order to assess student confidence and perceptions related to small-animal dentistry practice and related skills before and after the training. All results were compared between the two groups. Group A outperformed Group B in skill acquisition (*p* < 0.001) and there was greater improvement in skill confidence for Group A than Group B (*p* < 0.001). There was no statistical difference in perceptions related to small-animal dentistry between the two groups after the training (*p* ≥ 0.1). Group A rated their training experience more highly than Group B (*p* < 0.001). Although dental skill acquisition shows greater improvement when training is provided by models rather than video instruction, a blended approach to teaching dental skills is likely to be the best approach to optimizing dental skill acquisition.

## 1. Introduction

Periodontitis is the most frequently diagnosed disease in small-animal medicine [1], affecting over 87 percent of dogs over two years of age [2]. This is a disease that can cause bacteremia and which has been linked to several systemic diseases [3,4,5]. Pain associated with periodontitis and other dental diseases is well established in human patients and is one of the most common reasons patients seek dental treatment [6]. However, animals rarely display overt signs of oral pain associated with mild-to-moderate dental disease [7]. The lack of clinical signs in small-animal patients leads to dental disease progression and compromises health and welfare without the knowledge of even the most attentive of owners [7]. Therefore, the provision of dental services i.e., the prevention, diagnosis, and treatment of periodontitis and other dental diseases, is a paramount responsibility of the primary-care veterinary practitioner [8]. 

Unequivocally, small-animal dentistry knowledge and skills are highly sought by employers and are considered day-one competencies [8,9]. Small-animal practitioners have recognized the importance of new graduates having skills in dental prophylaxis, tooth extraction, and routine periodontal treatment [10]. New graduates are expected to perform these skills on an average of once per week with proficiency and minimal supervision [10]. However, the current curricula specific to veterinary dental education appear insufficient and limited. Specifically, a large survey of veterinary schools across the USA and the Caribbean found that, in pre-clinical years (years 1–3), small-animal veterinary dentistry was only taught as core course content in 30 percent of veterinary schools, and only as an elective in 23 percent of schools, while 17 percent of schools did not include it in the curriculum [8]. Furthermore, the same study reported that veterinary schools offered a mode of only one to four hours of lecture- and laboratory-based instruction, further illustrating the limited dentistry training opportunities provided to pre-clinical veterinary students [8]. The time students spend practicing dental skills in their clinical year (year 4) was unclear [8]. 

In recognizing the gap that exists between veterinary dental service demands and skills and in ensuring the adequacy of the small-animal dentistry skills of new graduates, there has been an increased focus in recent years on the teaching of small-animal dentistry to veterinary students. Outside of the lecture-based teaching of canine dentistry, the hands-on, deliberate practice of dental skills is fundamental to ensure graduates are practice-ready in this field [8,9]. In 2020, the teaching of dentistry was acknowledged as a requirement by the American Veterinary Medical Association (AVMA) [11]; this was further supported by the recommendations given by the World Small Animal Veterinary Association (WSAVA) [12]. 

To bridge the gap in dental skill instruction, models can be utilized to facilitate the acquisition of essential skills, such as those used to teach teeth cleaning skills, be they high-fidelity, mid-fidelity or low-fidelity [13]. Evidence supports the use of simulation as an essential part of modern medical curricula. Simulations are used to facilitate the acquisition of essential clinical skills and competencies through repetition and feedback within currently accepted learning theory frameworks. Mannequins have been used in human dental simulation training since the 1960s [14]. More recently, 3D-printed model teeth manufactured for use in simulation were found to be realistic and cost-effective [15]. A recent publication assessed student perceptions of four different modalities used to teach anatomy: natural teeth, 3D-printed teeth, a 3D virtual model and augmented reality (AR) [16]. The results showed that the natural teeth were of the highest educational value, the 3D-printed teeth were the easiest to use, and the AR model was the most interesting. However, the AR model scored the lowest rating for ease of use and educational value [16]. The paper concluded that there are limitations associated with AR, but that it is an area of significant ongoing development with potential for use in future dental training [16]. This example underscores that, in order to understand the benefits of models, validation of their use is essential, regardless of the level of fidelity, before their implementation in a curriculum. 

As advances are made in simulation development, the impact on trainee skill acquisition, or competency, must also be studied in order to provide an understanding of training outcomes. The assessment of competency is essential from both an educational and a financial perspective when the costs associated with model development and the management of simulation laboratories are considered. The human dental field is far more advanced than the veterinary field, with the use of haptic and virtual reality (VR) simulations having transformed the modern dental world [17]. While recognizing that additional studies are needed to further evaluate the use of VR systems in dental skill acquisition, it is important to acknowledge that VR simulators can collect, summarize and analyze all of an operator’s work to provide an objective assessment with real-time feedback, and that this assessment can be considered to be more appropriate than human expert assessment [17]. However, faculty feedback provides insight into weaknesses in student technique and provides context to procedural errors; more studies are needed to further evaluate the use of VR systems used in dental skill acquisition [18,19]. 

In contrast to the human field, the use of only a small number of dental models has been validated in veterinary medical education. A rudimentary model has been found to be an effective way of teaching dental cleaning when compared to the use of video instruction [20]. This same study also reported that all participating students agreed that training on a model would be beneficial to their skill acquisition and that all those who learnt using the model acknowledged improvements in their confidence [20]. In 2021, a study compared the use of three modes in teaching and practicing dental cleaning [13]. This study found that low-fidelity models are as effective as mid-fidelity and high-fidelity models for teaching cleaning. However, students were more accepting of the higher-fidelity models in the study [13]. Although low-fidelity models performed well in terms of instruction, experts recommended the use of higher-fidelity models for skill assessment [13]. In 2022, a published study investigated the use of 3D models to teach scaling and dental charting and found that students gained more confidence if models were used in advance of a cadaver laboratory [21]. 

There is no uniformity in how useful alternate fidelity simulation modalities are in terms of ensuring skill acquisition. As with veterinary dental skills, the positive effectiveness of low-fidelity models has been reported for veterinary surgical skills [22,23] and in human dentistry [24]. The appropriate validation and assessment of the use of veterinary models is essential; this is especially true in the field of small-animal veterinary dentistry, given the minimal available literature on this subject. 

The aim of this study was to evaluate the use of a high-fidelity model (HFM) and video instruction to teach three canine dental core skills to pre-clinical veterinary students: radiographic positioning; cleaning (scaling and polishing); and extractions. Students’ skill acquisition was assessed using an objective standardized clinical examination (OSCE) and students’ performance was compared between the two groups. Questionnaires were utilized to assess potential changes in the confidence of the students and their perceptions related to small-animal dentistry before and after receiving the training. The results were compared between the two groups. We adopted the hypothesis that students who practiced skills using different modalities, whether HFM or video instruction, would perform differently in the OSCE and have differing levels of confidence when applying skills. A secondary hypothesis was that exposing students to a novel modality, i.e., HFMs, would change their perception of and interest in small-animal dentistry. 

## 2. Materials and Methods

This study was conducted at the Ross University School of Veterinary Medicine (RUSVM), an American Veterinary Medical Association (AVMA)-accredited international veterinary school that provides an intensive pre-clinical course of seven semesters over two years and four months. RUSVM students complete their one year (three semesters) of clinical training at an affiliated AVMA-accredited veterinary school. 

At the time this study was conducted, small-animal dentistry instruction at RUSVM was delivered in (i) eight hours of didactic lectures in semester five, (ii) a one-hour self-directed online laboratory in semester six, and (iii) a three-hour elective canine live-animal laboratory in semester seven. The didactic lectures covered the pathology, diagnosis, treatment and prevention of common dental diseases and the core skills of radiology, cleaning, and extractions. The online laboratory focused on instrument identification and use, as well as the charting, and recognition and grading of pathology. The live-animal elective included dental radiography interpretation, probing, charting, and cleaning, with students performing supragingival and subgingival scaling and the polishing of all teeth in one quadrant of a dog’s mouth. 

Students in semester seven (third year) were invited to participate in this study. Participants were excluded if they were certified veterinary technicians (CVTs) (or international equivalent qualification) or if they had already participated in the seventh-semester elective live-animal laboratory. The study was conducted between January 2020 and January 2021 with three cohorts of students, with each cohort drawn from a different seventh semester. Over the duration of the three cohorts, 105 students enrolled in this study. An administrative colleague independent of the study applied the RAND formula in Microsoft Excel™ to a list of participating students for each cohort to randomly assign them into one of two instructional groups: one group trained using HFMs in a laboratory setting (group A), *n* = 52, and one group trained via video instruction (group B), *n* = 53. All study participants completed a written consent form. This study was approved by RUSVM’s Institutional Review Board (IRB), protocol number: 19-03-XP. 

The primary author developed a concise skill sheet, which was reviewed and approved by a Diplomate of the American Veterinary Dental College (AVDC) (fourth author). The skill sheet described (i) dental anatomy; (ii) equipment, instruments, and consumables; (iii) personal protective equipment (PPE); and (iv) the three skill sets: radiographic positioning, cleaning (scaling and polishing), and extractions. All participating students within each cohort received a copy of a skill sheet by email on the same day, providing them with a resource to complement their training. The skill sheet was used as the script in the video instruction and HFM laboratories and participants were encouraged to read it as many times as they preferred prior to watching the video or attending the HFM laboratory, as well as in advance of the OSCE. Additionally, the skill sheet was sent to the video instruction group in an email with a link to the video; these students were also informed that they could watch the video as many times as they felt appropriate. The HFM laboratories took place within one to four days of sending the skill sheet. All participants completed a dental skills OSCE two weeks after the HFM laboratories.

The 38 min video included a canine skull to present anatomy and a cadaverous head to demonstrate the skills, describing each element as they were presented or performed (Figure 1). 

The HFMs used in this study were made of silicone, plastic, and rubber. They were manufactured by Veterinary Simulators Industries Limited (VSI) (Figure 2). The HFMs had lips that could be reflected, as well as a tongue, an epiglottis, an esophagus, and a trachea. The maxillae and mandibles could be replaced with simulated teeth, bone, and soft rubber gingiva. The HFMs were provided with an arm and clamp that could be employed to secure the model to a tabletop, but these were not used in this study. 

At the time the study was conducted, three sets of jaws were available for use to teach different skillsets. (i) One set was used to teach scaling and polishing that included simulated calculus painted onto the teeth. Once dry, an ultrasonic scaler could be used to remove the simulated calculus as in a live patient. (ii) A second set with radiodense teeth was used to teach radiographic positioning (Figure 3). (iii) A third set was used to teach extractions, which included a simulated periodontal ligament. These techniques were used to perform gingival flap creation, alveolar bone removal, tooth elevation and tooth extraction (Figure 4). Components were colored in a realistic fashion to allow for the identification of different simulated tissues. 

This study asked three veterinarians to provide face validation for the models prior to their use in laboratories. We asked two diplomates of AVDC (one of them being the fourth author) and an experienced primary care clinician with an interest in small-animal veterinary dentistry (fifth author) to participate. They identified that the teeth of the radiology models had poor contrast. In response, VSI improved the contrast of the models. A subsequent assessment by the same veterinarians identified that the improved models included a variety of teeth that were found not to be strictly anatomically correct, and that some tooth roots had hollow appearances. However, the models were deemed to be appropriate for use in teaching radiographic positioning. In conclusion, the models were deemed to be appropriate for use to practice cleaning, radiographic positioning, and extraction skills, and were considered especially useful for learning instrument-handling skills and gaining familiarity with oral anatomy. 

One veterinarian (first author), one CVT (sixth author) and one certified veterinary technologist (seventh author) facilitated the training in the HFM laboratory and used the same script as that used in the video to present and guide the students through the skills. No additional information was provided to the students. All equipment, instruments and consumables used in the video were identical to those used in all laboratories and assessments. The HFM laboratories were delivered over three one-hour stations, one for each skill set: radiographic positioning; cleaning (scaling and polishing); and extraction. Small groups of students, *n* = 3–4, rotated through the stations. The HFM laboratories engaged students in performing the skills under supervision, receiving in-the-moment trainer feedback. The trainers in the HFM laboratories were the same individuals who developed the video and they were consistent in the delivery of content through all laboratories and cohorts.

Skill acquisition was assessed using a five-station, 13-item OSCE, and skills were assessed using a five-point Likert scale checklist: 1: below expectations, 2: novice, 3: advanced beginner, 4: competent, 5: proficient (Section A.1). The rubric was developed by the primary author and reviewed by the third author, who had significant experience in OSCE design, as well as by a diplomate of AVDC (fourth author). All stations were allocated 5 min to complete the tasks, with a 30 s inter-station change set aside.

Veterinarians and CVTs (or international equivalent) who were independent of the study rated student performance. Rater recruitment required individuals to have a minimum of five years of professional small-animal practice experience. Limited rater availability required the recruitment of raters of varied small-animal dental experience. Raters were provided with detailed guidance about their station and checklist prior to the start of the OSCE. Students were randomly scheduled for their OSCE times using Excel™ and raters were blinded to the training each student had received. 

Participating students completed online pre-training and post-OSCE questionnaires to assess their dental skill confidence and their perceptions related to small-animal dentistry (Section A.2 and Section A.3). The questionnaires consisted of 10 questions and required input on the basis of five-point Likert-scale response categories (e.g., not at all confident, slightly confident, somewhat confident, moderately confident, extremely confident). The pre-training questionnaire was completed before any training took place and before the skill sheet was shared. Pre-training questionnaires also included two binary response exclusion questions regarding participant qualifications and participation in the semester seven live-animal laboratory, as discussed earlier. The post-OSCE questionnaire included a Likert-scale response question regarding students’ opinions of the training they had received and an open-ended question asking students to provide feedback on the experience of the training they had received. Questionnaire answers were anonymous and collected through the Qualtrics online survey tool to ensure data protection.

All descriptive statistics include median and interquartile ranges. The Wilcoxon rank-sum test was used to compare each OSCE item score and the overall OSCE score between the two groups. Cronbach’s α was calculated in order to determine OSCE reliability. The impact of individual stations and items on the reliability of the OSCE was evaluated by comparing Cronbach’s α with or without each station or item.

To test the initial randomization, responses of the pre-training questionnaire were compared between the two groups using a Wilcoxon rank-sum test. The responses of the pre-training questionnaire were compared with the post-OSCE questionnaire using the Wilcoxon signed-rank test (paired by student) in order to determine if student confidence and perceptions improved throughout the study in both groups. Furthermore, the difference (subtraction) in the scores between pre-training and post-OSCE questionnaires was compared. In particular, we used the Wilcoxon rank-sum test to assess the level of change in confidence and perceptions between the two groups. Differences were considered significant for *p* value ≤ 0.05.

## 3. Results

Out of a total of 105 students, 52 attended the HFM laboratory, 53 completed the video instruction, and all 105 completed the OSCE. Student participation took place in January 2020 (*n* = 18); October 2020 (*n* = 21); and January 2021 (*n* = 66). There was no statistical difference in the baseline confidence or perceptions between the two groups before any training had taken place, *p* ≥ 0.2, confirming the random selection of groups (Section A.4).

Overall, group A outperformed group B, *p* < 0.001 (Table 1). Group A significantly outperformed group B for 7 of the 13 OSCE items. Group B did not outperform group A for any OSCE item. Station 4 reported that group A outperformed group B by the greatest degree, and Station 4 was the only station for which all items in a station produced a statistically significant difference in performance between the groups.

The OSCE had an overall reliability of α = 0.66 (95% confidence interval: 0.64–0.69) (Section A.5). Reliability varied significantly between stations, α = 0.003–0.59, and between raters, α = < 0–0.85 (Section A.5). Some items have more impact on reliability than others, and the overall OSCE reliability would increase, α > 0.66, if these items were removed from the analysis. This is true for all items in Station 1 and Item 1 in Station 3 (Section A.6).

A total of 84 students completed the pre-training and post-OSCE questionnaires, with 40 drawn from group A and 44 from group B. Group A had significantly greater confidence post-OSCE than pre-training in all six related questions, *p* < 0.001 (Section A.4). Group B had significantly greater confidence post-OSCE than pre-training in five of the six related questions, *p* ≤0.01 (Section A.4). There was no statistically significant improvement for any of the perception-related questions for either group, *p* ≥ 0.1 (Section A.4).

Group A had significantly greater improvements in confidence than the group B after the training and OSCE, with *p* < 0.001 for all related questions (Section A.4). In the post-OSCE questionnaire, group A rated their training experience to be better than group B’s, *p* < 0.001 (Section A.4). A total of 63 participants provided comments on their training experience in the post-OSCE questionnaire, *n* = 38 from group A, and *n* = 25 from group B. The primary author has summarized and purposely selected some of these comments in Table 2 to provide an overview of the students’ experience. 

## 4. Discussion

To the authors’ knowledge, this is the first published work that studies veterinary dental radiographic positioning and tooth extraction skill acquisition. In this study, the use of the HFMs to teach dental skills was shown to be of significant benefit when compared to video instruction, with the HFM group outperforming the video instruction group in the OSCE. Both video instruction and HFM laboratories improved skill confidence. However, there was a significantly greater increase for the HFM laboratory group, and the HFMs were rated as being better resources than the video. Neither training experience significantly affected perceptions. 

The positive impact of video resources on enhancing teaching and skill acquisition in human medical and dental education, as well as veterinary education, is widely reported [25,26,27,28,29], with the results of this study further supporting this. Although the HFM group outperformed the video instruction group in the OSCE, there was no statistical difference in improvement between the two groups for 6 of the 14 items. In total, 3 of these items were related to radiology skills, 2 items were related to cleaning, and 1 item was related to extractions. There are several possible explanations for the similarity in the acquisition of these skills between the two groups. Firstly, the complexity of these skills may have been such that the hands-on practice with HFM instruction was insignificant in terms of improving skill acquisition when compared to video instruction. Specifically, these skills could be considered easy to learn and can be taught through observation using video instruction. Secondly, students may have drawn from skills learnt elsewhere in the curriculum such as radiology laboratories and lectures, therefore bringing bias to the analysis. Thirdly, the potential of the HFMs may not have been realized due to an absence of the opportunity for deliberate practice, acknowledging that the use of deliberate practice in simulation-based medical education increases the success of clinical skill acquisition [30,31,32]. It is reasonable to surmise that the performance of the skills in question could have been influenced by the ease of developing an understanding via video instruction and the absence of opportunity for deliberate practice with the HFMs. The authors hypothesize that both video instructions and HFMs are valuable resources to teach dental veterinary skills and the level of difficulty can guide the approach. Varied teaching strategies can be employed to improve learning experiences, which is of particular interest to disciplines that may struggle to garner interest from students. A recent study investigated the use of a flipped classroom and peer-assisted learning to teach equine nutrition, a subject which most post-graduation veterinarians lack knowledge of and skills in [33]. Despite half of the students questioned in the study reporting to be uninterested in the subject matter, most expressed positive attitudes towards the teaching methods used [33]. Providing diverse learning opportunities has the potential to engage students with unpopular disciplines, which could raise the profile of under-resourced areas of study and ultimately improve animal welfare. 

The results of the post-OSCE questionnaire report that there was no significant overall increase in practical experience for the video instruction group; this is to be expected, as these students were not provided with any practical experience. However, the video did significantly improve confidence for all specific skill sets, and the OSCE results supported the notion that the video had a positive impact on skill acquisition. Although the HFM lab experience was rated higher than the video instruction in terms of improving skill acquisition, the video was also favorably rated. This result aligns with other research reports that find the use of video resources to be beneficial in terms of improving preparedness for assessments and reducing stress [34]. Students were less accepting of the notion that video instruction increased confidence in comparison with their relationship to HFM. This was potentially due to: (i) Students not spending sufficient time watching the video or doing so with due attention. Indeed, students reported varied, consumption of videos with peak viewing one or two days before assessment [34]. (ii) The length of the video being 38 min, considering that videos longer than 6 min have been found to have poor student engagement [35]. (iii) An absence of feedback and expert supervision, which are important in skill acquisition [22,36]. 

In the future, the authors recommend a blended approach to veterinary dental skill training, whereby videos lasting a maximum of 6 min are used in conjunction with the HFMs. Blended learning has well-documented benefits in terms of clinical skills development in the human and veterinary fields [37,38]. The use of technology provides students with the flexibility, convenience, and independence required for an individualized approach to learning [37,38]. The videos can be shared in advance of labs, used in lectures, and used as a resource for study outside of the classroom and laboratory. Veterinary students have been found to use online resources to prepare for practical classes [39]; further work on how veterinary students use videos to supplement their dental skill training would be beneficial to efforts to guide their use.

The HFM group in this study reported a greater improvement in confidence post-OSCE than the video instruction group. Indeed, an increase in confidence with skills learned using HFMs over lower-fidelity models is well reported in medical education [39,40,41]. Feedback and individualized learning with the active participation of the trainee are also very important to the success of simulation skill labs and are accepted in improving outcomes [39,40]. When considering the outcomes of this study, it is reasonable to conclude that the greater increase in confidence for the HFM group can be attributed to both the fidelity of the models and the training group size, *N* = 3–4, which allows for students to receive real-time individualized feedback from both instructors and peers.

Neither training experience significantly affected perceptions. Students’ backgrounds and interests in career choice have been reported positive influences on the outcome of associated clinical skill acquisition [42]. If an interest in a chosen career path is associated with success in clinical skill acquisition, it could also reinforce perceptions of skills, hence the lack of significant change from pre-training to post-OSCE.

This study does not consider other knowledge and skills students may be gaining when using the HFMs such as anatomy, patient handling, and personal posture, and practicing skills with water and a tongue in the simulated patient’s mouth. Simulation enhances psychomotor skills and hand–eye coordination, which are especially important for surgery [43]. Some of these experiences are difficult to achieve with lower-fidelity models, which is another area worthy of future consideration and study. The classification of simulation model fidelity is subjective and encompasses an array of features [39,44]; when compared with other veterinary models documented in the published literature, the authors consider their classification as high to be reasonable and in agreement with another published study [13]. However, if they are compared with models used in the human field, this would not be the case [18]. The lack of a standardized method of classification of simulated models creates challenges when comparisons and validation are sought [39]. Further consideration should be given to standardizing the classification of model fidelity between and within professions.

The overall reliability of the OSCE would be considered low, indicating differences in rating between raters. Raters A and E were CVTs with considerable experience of small-animal dentistry practice, and rater B was a CVT with little experience of small-animal dentistry. No CVT raters had experience of OSCE rating. Raters C, F, G and H were veterinarians with considerable experience of small-animal dentistry practice and rater D was a veterinarian with no experience of small-animal dentistry practice. All veterinarian raters had experience of OSCE rating. Reliability as raters varied considerably between raters and stations. However, it is notable that the raters with α < 0 were the CVT and the veterinarian with no significant small-animal dentistry practice experience. The results suggest that CVTs and veterinarians can assess pre-clinical veterinary students’ dentistry skills with a similar degree of reliability, and similarly, that both CVTs and veterinarians should be recruited as raters based on sound clinical dentistry experience and be provided with robust training in advance of performing an OSCE. Furthermore, acknowledging that rubric wording is important and is reported to impact assessors and pass/fail decisions [45], validity could be improved by increasing expert input to OSCE station and rubric design, robust rater training, and running pilots of the OSCE and rubric. There is additionally value in analyzing the results of specialists, experienced practitioners and new graduates whose skills have been assessed using the OSCE and rubric and comparing them with student results [46,47]. With investment in OSCE and rubric design and by analyzing the assessment results of different groups, over time the items could be refined and the reliability of these assessments improved. 

There are significant costs associated with the use of HFMs. Radiology jaw sets can be used with minimal wear over time but do eventually need replacement. The teeth of the models used for cleaning wear down after several uses and the cleaning procedures also cause wear to the model heads. Therefore, these models have a higher rate of turnover than radiology sets. The teeth of the extraction jaw sets cannot be replaced, each tooth is single-use, and extraction procedures cause wear to the model heads, meaning that judicious use of these models is warranted. In the HFM laboratories used for this study, each student extracted one incisor and one two-rooted tooth. The availability and storage of cadavers and other means of teaching, as well as budgets, will influence investment in the HFMs. Simulation models can only be as good as the educational environment in which they are used [44]. Therefore, these HFMs should be implemented into a curriculum with careful planning, and further assessment of their use is warranted. The models can also be used to teach nerve blocks and endotracheal intubation. Future positive validation of their use to teach these skills may influence laboratory investment.

## 5. Study Limitations

This study presents several limitations. Raters were assigned based on their availability. As such, there were limitations in maintaining rater–station continuity. Indeed, examiner cohorts are reported to have a significant and replicable influence on OSCE scores [48]. There are numerous possible factors that could be influencing the differences in rater assessment: rater experience of practicing small-animal dentistry, rater experience of assessing small-animal dentistry and rater experience of assessing the use of an OSCE. Additionally, numerous types of rater errors are recognized in medical assessments. Some of these errors are the most significant threats to assessment validity [49,50]. In the human dental field, rater training has been found to improve the inter-rater reliability of performance assessment, with rater training workshops and encouraging rater involvement with OSCE development used to facilitate these improvements [46,51]. Similarly, rater training has been recommended as a means to improve assessment validity in veterinary education [50]. Rater training could have been used in this study to improve reliability. However, rater impact is unlikely to have altered the results as they were blinded to the experimental group (HFM or video instruction) and participant order in the OSCE was randomly assigned. 

Students in the video instruction group could not have been expected to safely use a high-speed handpiece in the OSCE as they had not received in-person training to ensure that each student fully understood the necessary health and safety requirements of this handpiece, thus limiting the assessment of associated extraction skills. Further assessment of the acquisition of extraction skills is necessary.

## 6. Conclusions

Students greatly appreciated the opportunity to learn on HFMs and core dental skill acquisition showed greater improvement when training was provided with HFMs than video instruction. The use of HFMs to teach dentistry skills did not change students’ preference for small-animal dentistry over video instruction. The authors recommend the use of small-group simulation laboratory learning in rotation to facilitate the acquisition of multiple skills, this can be carried out for large cohorts if planned well. The primary author also recommends using appropriately trained and experienced CVTs as instructors in some clinical skills. CVTs and veterinarians can assess pre-clinical veterinary students’ dentistry skills with a similar degree of reliability. CVTs and veterinarians should be recruited based on sound clinical dentistry experience. Further work to assess the use of HFMs in a blended learning course, in comparison to cadavers and lower-fidelity models, and the transferability of acquired skills to the live patient should be conducted. In the future, a dental skills course that includes the use of videos lasting no longer than 6 min, and models of varied fidelity, may achieve similar outcomes to a course using HFMs alone and be preferable from a budgetary standpoint. 

## Figures and Tables

**Figure 1 vetsci-10-00526-f001:**
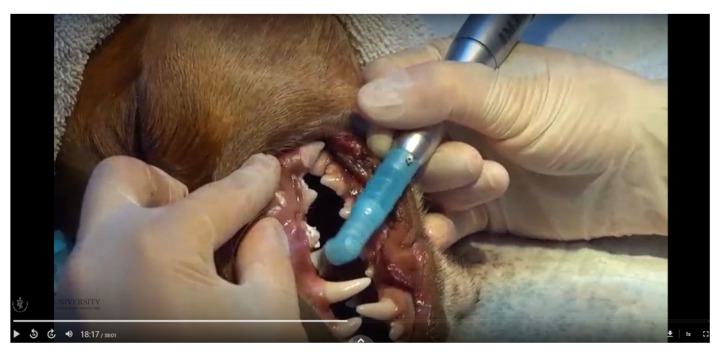
Video Screenshot Demonstrating Polishing.

**Figure 2 vetsci-10-00526-f002:**
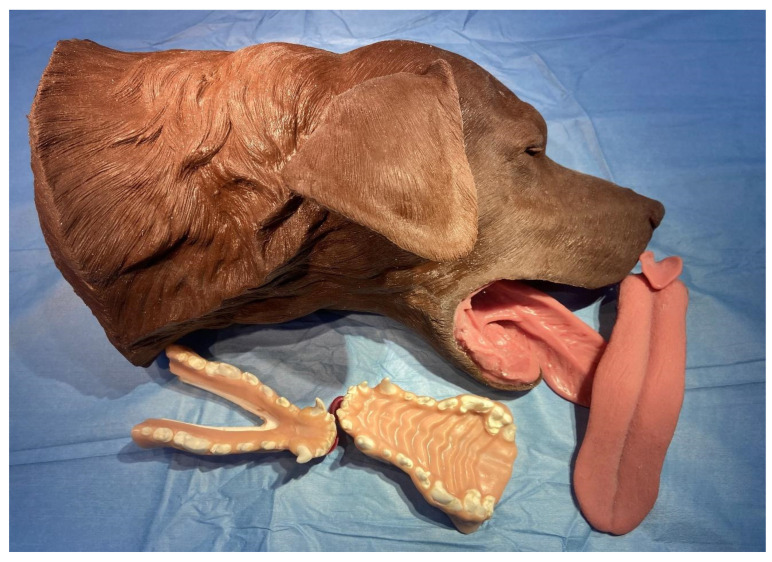
Constituent Parts of the High-Fidelity Model (HFM): head, maxilla, mandible, and tongue.

**Figure 3 vetsci-10-00526-f003:**
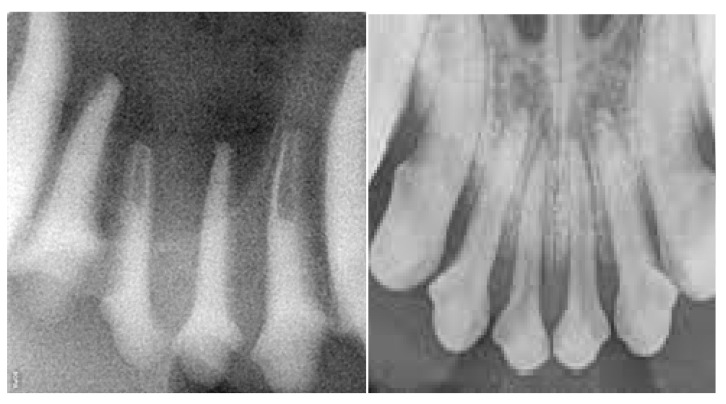
Radiographs of the Maxillary Incisors of the HFM (**left**) and a Live Patient (**right**).

**Figure 4 vetsci-10-00526-f004:**
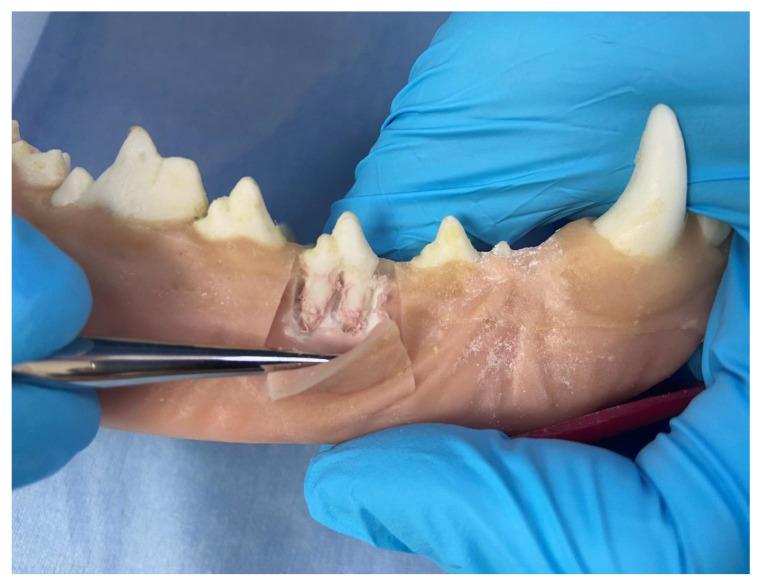
HFM Mandible Demonstrating Gingival Flap and Surgical Extraction Technique.

**Table 1 vetsci-10-00526-t001:** Comparison of OSCE Results Between Video Instruction and HFM Laboratory Groups.

	Grade HFM Laboratory Group A	Grade Video Instruction Group B	Wilcoxon Rank-Sum Test *p* Value
	Median (IQR)	Median (IQR)	
**Overall OSCE (denominator = 100)**	80.6 (72.4–85.8)	67.1 (56.7–71.3)	<0.001 *
**Station 1**			
Put on the necessary personal protective equipment to obtain dental radiographs.	4 (3–4)	2 (2–4)	0.008 *
Prepare the digital radiography sensor to obtain dental radiographs.	4 (2–4)	3 (2–4)	0.2
Place the canine head on the table and in the correct position to obtain a dental radiograph of the mandible. Indicate verbally to the examiner once you have completed this task.	3 (1–4)	2 (1–4)	0.6
**Station 2**			
Position the radiography sensor and the tube head of the radiograph generator to obtain a radiograph of the mandibular incisors at 45°. Indicate verbally to the examiner once you have completed the task.	3 (2–4)	3 (2–4)	0.8
Position the radiography sensor and the tube head of the radiograph generator to obtain a radiograph of mandibular premolar tooth 4 and molar tooth 1 in quadrant 3. Indicate verbally to the examiner once you have completed the task.	2 (1–3)	1 (1–2)	0.006 *
**Station 3**			
Put on the necessary personal protective equipment to perform a canine cleaning and polishing procedure.	4 (4–4)	4 (3–4)	0.3
Switch on the ultrasonic scaler machine and verify that it is working correctly.	4 (2–4)	2 (2–3)	<0.001 *
Switch on the air-driven dental unit and verify the three-in-one syringe handpiece is working correctly.	4 (4–4)	4 (4–4)	0.5
**Station 4**			
All teeth in the cadaver head have had all tartar removed. Demonstrate a scaling procedure on the left mandibular canine tooth.	3.5 (3–4)	2 (2–4)	0.001 *
Assemble the polisher handpiece so that it is ready for use and then polish the left mandibular canine tooth.	4 (3–4)	3 (2–4)	0.003 *
**Station 5**			
Select the necessary items to perform an incisor extraction. Place on Card A.	3.5 (3–4)	3 (2–4)	0.052
Select any elevator and use it to demonstrate how to hold this tool correctly when in use.	4 (4–4)	3 (2–4)	<0.001 *
Select the necessary instruments to perform a gingival flap—instruments for suturing are not necessary. Place on Card B.	3 (1–4)	1 (1–3)	0.03 *
Assemble the high-speed handpiece so that it is ready to remove alveolar bone from around a tooth root.	4 (3.75–4)	3 (1–3)	<0.001 *

IQR: interquartile range; grade for item: 1–5. * Statistically significant *p* values. Each item was scored using a five-point Likert scale checklist.

**Table 2 vetsci-10-00526-t002:** Selection of Participant Comments.

Comments from Group A
The high-fidelity models were really nice to learn on. I enjoyed getting the feel of how it is actually done on a live patient. I am amazed at what stuck with me after learning about it once for three hours two weeks ago.
As someone who has never been exposed to dentistry, I believe the lab helped me out a tremendous amount, I feel much more confident than I did prior to the in-person labs. I hope every vet student gets the chance to learn using the models, my confidence in dentistry has significantly improved after only one day of training.
I really enjoyed the models, I felt like it was a really good experience to feel and handle a model that’s similar to real life. The models were very helpful for getting a good understanding of accurate canine dental anatomy.
The high-fidelity models were extremely helpful for me because I could visualize and assess all procedures with all my senses. It was a great learning experience. I got very interested in dentistry because of the models.
**Comments from Group B**
Being assigned to the videos, I like the fact that I could review the material prior to the OSCE and rewatch them. However, I don’t think they are as helpful as a hands-on lab. There are many questions I wish I could have asked and even with watching the videos multiple times I don’t feel prepared to perform these skills on a live patient by myself.
I felt like I learned a lot watching the videos but not nearly as much as if I had done the lab in person. It’s hard to assess your understanding without being able to do something with direct instruction and immediate feedback.
I did not find the videos very helpful in learning how to perform the tasks demonstrated. However, I do feel that they would be a good to refer to after completing the lab on the models, should any questions come up after the lab.
Felt videos were good quality. However, online was able to slack off. There is also no feedback or possible corrections made to your skills with online videos and no live instructor.

^1^ Wilcoxon Signed-Rank Test (paired). ^2^ Wilcoxon Rank-Sum Test. * Statistically significant *p* values. *N* = 84.

## Data Availability

All data is contained within this article or Appendix A.

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
