# Peer review of "The Evaluation of a High-Fidelity Simulation Model and Video Instruction Used to Teach Canine Dental Skills to Pre-Clinical Veterinary Students"

_vetsci, 2023, doi:10.3390/vetsci10080526_

Round 1

Reviewer 1 Report

GENERAL COMMENTS

This is an interesting manuscript. A well presented study and easy to read.

The design of the study is very well organised and results are clearly presented.

Discussion and conclusions are also clear.

SPECIFIC COMMENTS

Few minor comments may be interesting for the authors to consider, namely

- Lines 237:  Please consider to include some explanation about the stations here. For example you can make a reference to Table 1 or to the Appendix.

- Lines 409-413: I believe that this information could also be part of the materials and methods session (e.g. explaining the profile of eveluators and how they were selected) and of the results session (e.g. explaining how they perfomed).

I hope this the above remarks are helpful.

Congratulations for the nice work.

Author Response

Ross University School of Veterinary Medicine

Island Main Road

Basseterre

St. Kitts & Nevis

West Indies

Aug 8th, 2023

Dear Editor and Reviewers,

Many thanks for your reviews of the manuscript, Evaluation of High-Fidelity Models and Video Instruction Used to Teach Small Animal Dental Skills to Pre-Clinical Veterinary Students, your feedback is extremely helpful, and your kind words are most appreciated.

The manuscript has been reviewed with your comments in mind and edits have been made where applicable. Please see my responses to your comments below. Edits are highlighted in the reviewed manuscript I have attached. Removal of highlights produces a clean copy.

I look forward to any further feedback you may provide.

Kind regards,

James Fairs.

Response to Reviewer 1 Comments

Lines 237:  Please consider to include some explanation about the stations here. For example, you can make a reference to Table 1 or to the Appendix.

  • Agreed, sentence expanded to include the specific skill sets on Line 240, The HFM laboratories were delivered over three one-hour stations, one for each skill set: radiographic positioning; cleaning (scaling and polishing); and extractions.’
  • I have not referenced Tables or Appendix as they contain details of OSCE stations, not the training laboratory stations.

Lines 409-413: I believe that this information could also be part of the materials and methods session (e.g. explaining the profile of evaluators and how they were selected) and of the results session (e.g. explaining how they performed).

  • Agree to the comment about Materials and Methods, Edits made to Lines 252-254 in Materials and Methods, ‘Veterinarians and CVTs (or international equivalent) independent of the study rated student performance. Rater recruitment required individuals to have a minimum of five years of professional small animal practice experience. Limited rater availability required recruitment of raters of varied small animal dental experience.’
  • Regarding the comment about results: the performance of raters is stated in the Results section, Lines 305-307, ‘The OSCE had an overall reliability of α = 0.66 (95% Confidence Interval: 0.64 – 0.69) (Appendix A5). Reliability varied significantly between stations, α = 0.003 – 0.59, and between raters α = <0 – 0.85 (Appendix A5).’ These results are then discussed at length in the Discussion Lines 410-422. I would prefer these elements to remain unedited if the reviewers are in agreement.

Reviewer 2 Report

Dear Authors,

Congratulations for  the impressive work you have submitted. The research you've conducted is thorough, insightful, and contributes valuable knowledge to the field of dentistry education. The innovative exploration of high-fidelity models (HFMs) and the implications of their usage in a learning environment certainly make for an interesting read.
However, after a careful review, I believe the paper would benefit from minor revisions to further enhance its clarity and impact. Primarily, the manuscript could gain from a more defined structure, which includes subdividing the text into smaller, more digestible sections.In addition, I suggest that in your discussion section, it would be advantageous to draw comparisons with other innovative teaching methodologies in veterinary education. Specifically, you might want to reference a recent study that explores the use of a flipped classroom and peer-assisted learning approach in teaching equine nutrition, despite initial student disinterest ( https://doi.org/10.1016/j.jevs.2023.104537  ). Adding this comparison could provide a broader context for your readers about the diversity of successful teaching strategies within veterinary education.Overall, these are minor modifications and do not detract from the quality and significance of your research.

Author Response

Ross University School of Veterinary Medicine

Island Main Road

Basseterre

St. Kitts & Nevis

West Indies

Aug 8th, 2023

Dear Reviewer,

Many thanks for your reviews of the manuscript, Evaluation of High-Fidelity Models and Video Instruction Used to Teach Small Animal Dental Skills to Pre-Clinical Veterinary Students, your feedback is extremely helpful, and your kind words are most appreciated.

The manuscript has been reviewed with your comments in mind and edits have been made where applicable. Please see my responses to your comments below. Edits are highlighted in the reviewed manuscript I have attached. Removal of highlights produces a clean copy.

I look forward to any further feedback you may provide.

Kind regards,

James Fairs.

Response to Reviewer 2 Comments

  • Consider replacing "most diagnosed" with "most frequently diagnosed" for improved clarity.
    • Line 47, edit made as above.
  • Rephrase the sentence to improve readability, for instance: "However, the current curricula specific to veterinary dental education appears insufficient and limited."
    • Lines 63-64, edit mad as above.
  • It might be more informative to specify what kind of "models" you're referring to here.
    • This is on Line 82, on Line 83 as part of the same sentence, statement added, ‘…such as those used to teach teeth cleaning skills…
  • Consider breaking this long sentence into two for improved readability.
    • Lines 92-95. I have considered this, but I think it reads better as it is. I would prefer to leave this unedited if the reviewers agree.
  • This sentence could be restructured for clarity.
    • Lines 100-104 edited as follows: ‘As advances are made in simulation development, the impact on trainee skills acquisition, or competency, must also be studied to provide understanding of training outcomes. The assessment of competency is essential from both an educational and a financial perspective when the costs associated with model development and the management of simulation laboratories are considered.’
  • Break up this sentence for clarity. Mention didactic lectures in semester five separately and then discuss the content of those lectures.
    • Lines 152-161, now edited to four sentences: ‘At the time this study was conducted, small animal dentistry instruction at RUSVM was delivered in (i) eight hours of didactic lectures in semester five, (ii) a one-hour self-directed online laboratory in semester six, and (iii) a three-hour elective canine live-animal laboratory in semester seven. The didactic lectures covered pathology, diagnosis, treatment and prevention of common dental diseases and the core skills of radiology, cleaning, and extractions. The online laboratory focused on instrument identification and their use, charting, and recognition and grading of pathology. The live-animal elective included dental radiography interpretation, probing, charting, and cleaning, with students performing supragingival and subgingival scaling and polishing of all teeth in one quadrant of a dog’s mouth.’
  • To avoid confusion, specify what the "cleaning procedure" entails.
    • Line 160-161 edited, see final sentence above.
  • Specify the method used for random assignment of students into instructional groups.
    • Lines 167-171 edited to reflect this: Over the time of the three cohorts, 105 students enrolled in this study. An administrative colleague independent of the study applied the RAND formula in Microsoft Excel™ to a list of participating students for each cohort to randomly assign them into one of two instructional groups: one group trained using HFMs in a laboratory setting (group A), n=52, and one group trained by video-instruction (group B), n=53.’
  • I suggest introducing a comparison to innovative methodologies in veterinary education at line 354, as your mention of the complexity of skills being taught through video instruction provides a nice segue to the related study: "It is reasonable to surmise that performance of the skills in question could have been influenced by ease of understanding by video instruction and the absence of opportunity for deliberate practice with the HFMs. The authors hypothesize that both video instructions and HFMs are valuable resources to teach dental veterinary skills and the level of difficulty can guide the approach. In this context, it is worth noting a recent study that highlights the successful implementation of a flipped classroom and peer-assisted learning in equine nutrition education, despite initial student disinterest [doi.org/10.1016/j.jevs.2023.104537]. This showcases the diversity and potential of various teaching strategies within the broader field of veterinary education." This adjustment can enrich the discussion section by setting your work within a larger educational framework and expanding upon its significance.
    • This raises a very interesting point which has been incorporated into the manuscript on Lines 356-366 as follows, ‘The authors hypothesize that both video instructions and HFMs are valuable resources to teach dental veterinary skills and the level of difficulty can guide the approach. Varied teaching strategies can be employed to improve learning experiences, this is of particular interest to disciplines which may struggle to garner interest from students. A recent study investigated the use of a flipped-classroom and peer-assisted learning used to teach equine nutrition, a subject which most graduated veterinarians lack knowledge and skills [34]. Despite half of the students questioned in the study reporting to be uninterested in the subject matter, most expressed positive attitudes towards the teaching methods [34]. Providing diverse learning opportunities has the potential to engage students with unpopular disciplines, this could raise the profile of under-resourced areas of study and ultimately improve animal welfare.’
    • The reference section has been fully updated to reflect this.
  • The manuscript could gain from a more defined structure, which includes subdividing the text into smaller, more digestible sections.
    • Within the stipulated guidelines for this journal further subdivisions/subtitles etc are challenging to implement. The editor’s feedback (next section below) has also been considered. An additional section, Study Limitations, has been added to Line 455. I hope this is sufficient to address this concern.
